# Yersiniabactin-Producing *E. coli* Induces the Pyroptosis of Intestinal Epithelial Cells via the NLRP3 Pathway and Promotes Gut Inflammation

**DOI:** 10.3390/ijms241411451

**Published:** 2023-07-14

**Authors:** Hao Wang, Chun-Lang Shan, Bin Gao, Jin-Long Xiao, Jue Shen, Jin-Gang Zhao, Dong-Mei Han, Bin-Xun Chen, Shuai Wang, Gen Liu, Ai-Guo Xin, Long-Bao Lv, Peng Xiao, Hong Gao

**Affiliations:** 1College of Food Science and Technology, Yunnan Agricultural University, Kunming 650201, China; wanghaoxu0117@163.com (H.W.); doc_gao@sina.com (B.G.); 2Faculty of Animal Science and Technology, Yunnan Agricultural University, Kunming 650201, China; 13614470751@163.com (C.-L.S.); zjingang2022@163.com (J.-G.Z.); 3College of Veterinary Medicine, Yunnan Agricultural University, Kunming 650201, China; xiaojl_h@163.com (J.-L.X.); nicetomeet_jue@163.com (J.S.); hdm18314447448@163.com (D.-M.H.); chenbingxun425@126.com (B.-X.C.); ws3031166768@163.com (S.W.); liugen2041640013@163.com (G.L.); 4National Foot-and-Mouth Disease Para-Reference Laboratory (Kunming), Yunnan Animal Science and Veterinary Institute, Kunming 650224, China; aiguo_xin@hotmail.com; 5National Resource Center for Non-Human Primates, National Research Facility for Phenotypic & Genetic Analysis of Model Animals (Primate Facility), Kunming Institute of Zoology, Chinese Academy of Sciences, Kunming 650107, China; lvlongbao@mail.kiz.ac.cn

**Keywords:** *E. coli*, Ybt HPI, pyroptosis, NLRP3, gut inflammation

## Abstract

The high-pathogenicity island (HPI) was initially identified in Yersinia and can be horizontally transferred to *Escherichia coli* to produce yersiniabactin (Ybt), which enhances the pathogenicity of *E. coli* by competing with the host for Fe^3+^. Pyroptosis is gasdermin-induced necrotic cell death. It involves the permeabilization of the cell membrane and is accompanied by an inflammatory response. It is still unclear whether Ybt HPI can cause intestinal epithelial cells to undergo pyroptosis and contribute to gut inflammation during *E. coli* infection. In this study, we infected intestinal epithelial cells of mice with *E. coli* ZB-1 and the Ybt-deficient strain ZB-1Δ*irp2*. Our findings demonstrate that Ybt-producing *E. coli* is more toxic and exacerbates gut inflammation during systemic infection. Mechanistically, our results suggest the involvement of the NLRP3/caspase-1/GSDMD pathway in *E. coli* infection. Ybt promotes the assembly and activation of the NLRP3 inflammasome, leading to GSDMD cleavage into GSDMD-N and promoting the pyroptosis of intestinal epithelial cells, ultimately aggravating gut inflammation. Notably, NLRP3 knockdown alleviated these phenomena, and the binding of free Ybt to NLRP3 may be the trigger. Overall, our results show that Ybt HPI enhances the pathogenicity of *E. coli* and induces pyroptosis via the NLRP3 pathway, which is a new mechanism through which *E. coli* promotes gut inflammation. Furthermore, we screened drugs targeting NLRP3 from an existing drug library, providing a list of potential drug candidates for the treatment of gut injury caused by *E. coli*.

## 1. Introduction

The pathogenicity island (HPI) found in Yersinia is capable of synthesizing, regulating, and transporting the siderophore yersiniabactin (Ybt) [1,2]. This island is only present in virulent strains [3]. The functional core region of HPI is formed via the *irp2–irp1–irp3–irp4–irp5–FyuA* gene axis, and mutations in *irp2* can prevent Ybt synthesis [4]. Although Ybt HPI is primarily found in Yersinia, it has been detected in *Escherichia coli* with a homology of 98–100% [5,6]. Ybt HPI enhances the pathogenicity of *E. coli* by competing with the host for Fe^3+^ [6,7,8]. Several studies have demonstrated that *E. coli* containing HPI can activate autophagy in host cells through Ybt [6,9], promote inflammatory responses, upregulate tumor necrosis factor-α(TNF-α) and interlenkin 1β (IL-1β) levels in pig small intestinal epithelial cells [10], and induce macrophage pyroptosis by promoting the expression of caspase-1 pathway-related factors [11]. However, it remains unclear whether Ybt can also induce pyroptosis of intestinal epithelial cells and thereby contribute to the gut inflammation caused by *E. coli* infection.

Pyroptosis is a novel form of programmed cell death that is mediated by gasdermin family proteins and was first discovered in macrophages [12,13,14]. Pyroptosis is characterized by nuclear condensation, as well as the cleavage and oligomerization of gasdermin family proteins’ amino-terminus, leading to the formation of cleavage pores in the plasma membrane [15]. In contrast to apoptosis, pyroptosis not only promotes DNA fragmentation but also results in the release of a large number of pro-inflammatory factors [16]. It can be divided into two pathways: the classical pathway, which is dependent on caspase-1, and the noncanonical pathway, which is dependent on caspase-4, 5, and 11 [17,18]. The classical inflammasomes that depend on caspase-1 are primarily classified as NLRP1, NLRC4, NLRP3, AIM2, and Pyrin inflammasomes [19]. These inflammasomes are activated by sensing specific signals through NOD-like receptors (NLRs) and recruit caspase-1 through adaptor proteins [20]. Upon recruitment, pro-caspase-1 undergoes self-cleavage, leading to the generation of active caspase-1. In the cytoplasm, the N-terminal and C-terminal regions of gasdermin D (GSDMD) interact to maintain an inhibitory state, and active caspase-1 triggers the cleavage of GSDMD, resulting in the release of its active form, GSDMD-N. GSDMD-N then binds to the cell membrane, forming pores known as GSDMD-N pores [13,21]. GSDMD-N pores release IL-1β and IL-18 into the extracellular space, thereby amplifying the inflammatory response [20,21,22].

The objective of this study was to investigate the contribution of Ybt HPI to the pathogenicity of *E. coli*. To this end, we identified a novel mechanism whereby Ybt triggers pyroptosis in intestinal epithelial cells, potentially exacerbating gut inflammation through the modulation of the NLRP3 pathway.

## 2. Results

### 2.1. Ybt Promotes Systemic Infection Caused by E. coli

The existence of Ybt HPI has been demonstrated to be associated with the pathogenicity of *E. coli* [10,23,24]. In this study, we established a mouse model of *E. coli* infection (Figure 1a) and evaluated the clinical score and body temperature of the infected mice. We observed that mice infected with Ybt-producing ZB-1 had severe injuries, while those infected with ZB-1Δ*irp2* had moderate injuries (Figure 1b). Furthermore, the body temperature of the infected mice reached its maximum 24 hours after infection (Figure 1c). Upon dissection, we found that Ybt-expressing ZB-1 caused significant organ damage, while the damage was reduced in mice infected with the ZB-1Δ*irp2* strain (Figure 1d). We also observed a significant rise in the levels of inflammatory factors, such as IL-18 and IL-1β, in the serum of infected mice, with the ZB-1 group showing significantly higher levels than the ZB-1Δ*irp2* group, peaking 24 hours after infection (Figure 1e,f).

Furthermore, based on the significant secretion of inflammatory factors IL-18 and IL-1β observed at the 24 h time point, we decided to conduct subsequent studies specifically at this time point. Pathological observation revealed that Ybt exacerbated liver and kidney damage induced by *E. coli*, resulting in massive tissue hemorrhage. TEM showed severe cell damage, massive accumulation of lipid droplets, and nuclear pyknosis, indicating that the cells were on the verge of death (Figure 1d). Moreover, Ybt promoted *E. coli*-induced organomegaly and significantly increased the spleen, kidney, and liver organ indices (Figure 1g). Collectively, these findings demonstrate that Ybt enhances *E. coli*-induced systemic infection in mice.

### 2.2. Ybt Promotes Gut Inflammation Induced by E. coli

It is well established that Ybt can promote *E. coli*-induced inflammation in intestinal epithelial cells [3,10]. To investigate the role of Ybt in *E. coli*-induced intestinal injury, we examined the intestinal tract of mice infected with *E. coli*. Compared with ZB-1Δ*irp2*, ZB-1 caused significant damage to the integrity of the intestinal villi and the intestinal wall (Figure 1d and Figure 2a,b). Furthermore, TEM showed that Ybt promoted *E. coli*-induced nuclear pyknosis and lipid droplet agglomeration (Figure 2a). Figure 2c illustrates that infection with Ybt-producing ZB-1 significantly increased the content of sIgA in the small intestinal mucus, indicating a strong stimulation of intestinal mucosal immunity by Ybt. Furthermore, we observed a decrease in SOD activity and the presence of LDH in the intestine, which are known markers of intestinal injury [25]. Our findings revealed that ZB-1 infection significantly aggravated intestinal damage, resulting in the release of LDH and the inhibition of SOD activity. However, these harmful effects were alleviated in the ZB-1Δ*irp2*-infected group (Figure 2d,e). We also observed a large accumulation of inflammatory cytokines IL-18 and IL-1β in intestinal epithelial cells, and ZB-1 infection significantly promoted this result (Figure 2a,f,g).

DNA fragmentation is a prerequisite for cell damage and death [26]. Our results showed that TUNEL positivity, similar to apoptosis, occurred in intestinal epithelial cells, and ZB-1 infection significantly promoted this result relative to the ZB-1Δ*irp2* group (Figure 2a,f). Overall, these data suggest that Ybt promotes *E. coli*-induced gut inflammation by inducing DNA fragmentation and the release of inflammatory factors.

### 2.3. E. coli-Ybt Induces Pyroptosis through the NLRP3 Pathway

It is well established that TUNEL-positive cells are characteristic of apoptosis [27], which does not typically induce an inflammatory reaction. This contradicts the fact that large quantities of IL-18 and IL-1β are released, suggesting that pyroptosis may be involved in *E. coli*-Ybt-induced intestinal injury. To test this hypothesis, we first detected the cleavage of GSDMD in the intestinal tract. Our results showed that infection with Ybt-producing ZB-1 significantly promoted the cleavage of GSDMD into GSDMD-N, similar to the pyroptosis-positive control LPS group, relative to the ZB-1Δ*irp2* group. Immunohistochemistry and Western blot analyses confirmed the abundant expression of GSDMD-N in intestinal epithelial cells (Figure 3a–d).

To further investigate the molecular mechanism underlying *E. coli*-induced pyroptosis, we reanalyzed the dataset GSE124917 [28] in the GEO database. Our results showed that NLRP3, caspase-1, GSDMD, and IL-1β were significantly upregulated in response to *E. coli* infection, with KEGG enrichment revealing that upregulated genes were mainly concentrated in the NOD-like receptor signaling pathway (Figure 3e–g). Protein interaction network analysis revealed that the pyroptosis classic pathway NLRP3/caspase-1/GSDMD is involved in *E. coli* infection (Figure 3h).

Next, we measured the mRNA levels of each gene in the NLRP3/caspase-1/GSDMD pathway after *E. coli* infection. Our results showed that infection with Ybt-producing ZB-1 significantly upregulated the mRNA levels compared with the ZB-1Δ*irp2* group (Figure 4a). We also observed the assembly of NLRP3 inflammasomes in the intestine, which was significantly promoted by *E. coli* infection, especially in the Ybt-producing ZB-1 group. Furthermore, we detected a large number of inflammatory factors IL-18 and IL-1β in the intestine, which were also significantly promoted by Ybt-producing ZB-1 (Figure 4b–e). In addition, IHC analysis revealed that the expressions of NLRP3 and caspase-1 were mainly concentrated in intestinal epithelial cells (Appendix A). These findings support our hypothesis that pyroptosis mediated by the NLRP3 pathway plays a critical role in *E. coli*-Ybt-induced gut inflammation.

### 2.4. E. coli-Ybt Induces Pyroptosis of Small Intestinal Epithelial Cells

Based on the results obtained, we found that GSDMD-N expression and NLRP3 inflammasome assembly were mainly localized in the enterocytes of the intestinal villi. To further investigate the effect of *E. coli*-Ybt on pyroptosis in intestinal epithelial cells, we studied the cell damage at different time points after infection (Figure 5a). Our findings showed that the most severe cell damage occurred at 6 h after *E. coli* infection, and the presence of Ybt-producing ZB-1 significantly increased LDH release (Figure 5b). Furthermore, we observed a significant increase in IL-18 and IL-1β release in the supernatant of cells infected with *E. coli*, which are characteristic indicators of pyroptosis (Figure 5c,d). We also investigated the mRNA levels of the genes involved in the NLRP3/caspase-1/GSDMD pathway in cells after *E. coli* infection and found that ZB-1 infection significantly upregulated their expression levels (Figure 5e–h). Moreover, the assembly of the NLRP3 inflammasome was found to be promoted in the ZB-1 infected cells, similar to that in the LPS group (Figure 5i). Importantly, we observed the activation of NLRP3 and the cleavage of GSDMD-N in ZB-1-infected IPE2-J2 cells (Figure 5j). In addition, HE staining revealed that ZB-1 infection promoted intestinal epithelial cell damage, resulting in cell swelling and nuclear condensation (Figure 5i). Collectively, these findings suggest that *E. coli*-Ybt can activate the assembly of the NLRP3 inflammasome, promote pyroptosis in intestinal epithelial cells, and induce cellular inflammation.

### 2.5. Knockdown of NLRP3 Alleviates E. coli-Ybt-Induced Pyroptosis

To further reveal the role of NLRP3-mediated pyroptosis in *E. coli*-Ybt-induced intestinal epithelial cell injury, we employed small interfering RNA (siRNA) targeting NLRP3 to knockdown NLRP3 expression in IPEC-J2 cells (Appendix A). The qPCR results showed that the upregulated mRNA levels (*ASC*, *GSDMD*, *caspase-1*, *IL-18*, and *IL-1β*) of *E. coli*-Ybt were significantly reduced after NLRP3 knockdown (Figure 6a). Furthermore, the colocalization results of NLRP3 and caspase-1 expression showed that the assembly of NLRP3 inflammasome promoted by *E. coli*-Ybt was alleviated after knocking down NLRP3 (Figure 6b). Subsequently, in cells infected with the ZB-1 strain, we observed a decrease in GSDMD-N expression after knocking down NLRP3 compared with IPEC-J2 cells (Figure 6c). This decrease in GSDMD-N expression resulted in a reduced release of inflammatory cytokines IL-18 and IL-1β (Figure 6d,e). Additionally, HE staining showed that inhibiting the expression of NLRP3 could alleviate the intestinal epithelial cell injury induced by *E. coli*-Ybt (Figure 6f).

GSDMD-N-mediated plasma membrane pores are another hallmark of pyroptosis [29]. Therefore, we used PI staining to detect plasma membrane pore formation and showed that ZB-1 infection significantly promoted plasma membrane pore formation relative to the ZB-1Δ*irp2* group, which was reduced after NLRP3 knockdown in IPEC-J2 cells (Figure 7a,b). Importantly, we observed the presence of plasma membrane pores via TEM (Appendix A). Given the role of NLRP3 in promoting the damage of *E. coli*-Ybt, we further predicted the interaction between Ybt and NLRP3, and CB-Dock2 predicted five better combination models (Vina scores are shown in Appendix A). The optimal results we selected showed that Ybt can be well combined with NLRP3 (Figure 7c–d), and the Vina score was −9.1, which supports our hypothesis. In summary, these data indicate that NLRP3-pathway-mediated pyroptosis is a crucial mechanism underlying *E. coli*-Ybt-induced cell damage.

### 2.6. Screening of Drug Libraries Targeting NLRP3

Given the critical role of NLRP3 in *E. coli* infection, we conducted a search in existing drug databases to identify the compounds that target NLRP3. After analysis, we selected five drug molecules that demonstrated the best binding affinity (drug information and Vina score can be found in Appendix A), which are digitoxin, eltrombopag, isavuconazonium, dexamethasone metasulfobenzoate, and paliperidone. The binding interactions between these drug molecules and NLRP3 are depicted in Figure 8a–e.

## 3. Discussion

The invasion of pathogenic *E. coli* into the body usually results in severe systemic infection, often accompanied by gut inflammation [30,31]. In this study, we demonstrated that Ybt-producing *E. coli* is more toxic, exacerbating gut inflammation during systemic infection. Furthermore, we investigated the molecular mechanism underlying the aggravation of gut inflammation by Ybt and found that NLRP3-pathway-induced pyroptosis is closely related to the gut inflammation caused by *E. coli*-Ybt. *E. coli*-Ybt promotes the assembly and activation of the NLRP3 inflammasome, cleaves GSDMD to GSDMD-N, promotes inflammation, and ultimately leads to pyroptosis. Importantly, NLRP3 knockdown alleviated the aforementioned phenomenon, and the binding of free Ybt to NLRP3 may be the trigger. Previous theories suggested that Ybt HPI enhances the pathogenicity of *E. coli* mainly by competing with the host for Fe^3+^ [7,8]. However, we unexpectedly found that *E. coli*-Ybt can induce pyroptosis in intestinal epithelial cells, promoting body damage. The NLRP3/caspase-1/GSDMD pathway is the key point of pyroptosis induced by *E. coli*-Ybt.

Pyroptosis is a highly pro-inflammatory programmed cell death that was first observed in macrophages after bacterial infection or bacterial toxin treatment [14,32]. Our previous study demonstrated that *E. coli* containing HPI can promote pyroptosis in macrophages [11]. In this study, we observed that Ybt promotes chromosome aggregation in intestinal epithelial cells caused by *E. coli* infection, resulting in a large number of cells being TUNEL positive, and a substantial increase in inflammatory IL-1β and IL-18 in intestinal epithelial cells. Initially, we believed that *E. coli*-Ybt promoted the occurrence of apoptosis, but since the cells were in immune quiescence when apoptosis occurred, this contradicts the fact that *E. coli*-Ybt triggers a more severe and intense intestinal inflammatory response. Therefore, we hypothesize that *E. coli*-Ybt may induce pyroptosis in intestinal epithelial cells. Firstly, we detected the cleavage of GSDMD-N, which is an executor of pyroptosis [29,33,34,35]. As expected, we observed the production of GSDMD-N, and *E. coli*-Ybt promoted the generation of cell membrane pores. To further elucidate the mechanism of pyroptosis induced by *E. coli* infection, we mined the dataset (GSE124917) in the GEO database and found that *E. coli* infection significantly upregulates NLRP3, caspase-1, GSDMD, and IL-1β. Interestingly, other pyroptosis pathway factors, such as NLRP1, 2, and 4; caspase-2, 3, 6, 8, and 9; and GSDMB and GSDMC were not significantly upregulated. Importantly, we observed in vivo and in vitro that *E. coli*-Ybt promotes the activation and assembly of NLRP3, which is similar to *E. coli*-induced epithelial cell pyroptosis [36], thereby verifying our hypothesis.

In the process of bacterial infection, pyroptosis can induce host cell death, which may be beneficial for the body to eliminate pathogenic bacteria and protect itself [32,37,38]. However, pyroptosis also has disadvantages, such as promoting the release of IL-1β and IL-18 and aggravating the inflammatory response [39,40,41]. Our study demonstrated that Ybt-induced *E. coli* infection led to intestinal epithelial cell pyroptosis via the NLRP3 pathway, resulting in the release of large amounts of IL-1β and IL-18, which promoted gut inflammation and contributed to *E. coli*-induced intestinal damage. To determine the crucial role of NLRP3, we knocked down NLRP3 expression in intestinal epithelial cells using specific siRNA and found that pyroptosis rates decreased, limiting *E. coli*-induced inflammatory responses, even when infected with Ybt-producing ZB-1. To further elucidate the mechanism through which *E. coli*-Ybt promotes NLRP3 inflammasome assembly, we hypothesize that free siderophore Ybt may be recognized and bound by NLRP3, thereby activating NLRP3 inflammasome assembly. Our docking results support this hypothesis, but further experimental verification is needed. Currently, the most effective way to treat bacterial infections is the use of antibiotics, but this practice can lead to the emergence of antibiotic-resistant superbugs. Given the role of NLRP3 in *E. coli* infection, we screened drugs targeting NLRP3 using a drug library and identified candidates that effectively bind to NLRP3 (e.g., digitoxin, a cardiotonic drug), providing a list of potential drugs for treating *E. coli*-induced gut inflammation in the future.

The complexity of regulatory signals within the body necessitates a more comprehensive investigation of the inflammasomes involved in *E. coli* infection. Although NLRP3 was observed to play a role in this process through the analysis of human datasets, we recognize that our focus on NLRP3 alone may not have been sufficiently rigorous. In our cell experiments, we found that knocking down NLRP3 with siRNA inhibited inflammasome assembly and pyroptosis. Additionally, molecular docking predicted the *E. coli*-Ybt pre-NLRP3 binding. However, to strengthen our findings, future studies could incorporate reverse validation using NLRP3 null mice. This would provide further evidence for the involvement of NLRP3 in the observed effects. Furthermore, Shi et al. identified human caspase-4/5 and mouse caspase-11 as intracellular receptors for bacterial lipopolysaccharide (LPS), leading to pyroptosis [42]. While our use of LPS as a positive control was not rigorous or accurate, it was initially chosen to serve as a strong inflammatory response control, highlighting the intestinal inflammation induced by *E. coli*-Ybt. We acknowledge the need for more precise control measures in future studies. Additionally, it is important to note that LPS is an essential component of the *Escherichia coli* cell wall. The study by Vanaja et al. (Cell, 2016) reported that bacterial outer membrane vesicles (OMVs) promote the cytosolic localization of LPS and the activation of caspase-11 [43]. This finding suggests that LPS present in OMVs could be a significant confounding factor influencing the conclusions of our study. To address this potential confounder, future research could employ gene editing technology to generate strains with reduced LPS OMV secretion. By conducting infection experiments using these modified strains, we can further validate our conclusions.

In conclusion, this study presents novel evidence that Ybt exacerbates host injury and underscores the critical role of the NLRP3 pathway in *E. coli*-induced gut inflammation (Figure 9). Our findings pave the way for further investigations into innovative therapies targeting the Ybt-NLRP3 axis to alleviate *E. coli*-induced gut inflammation. Future research will aim to elucidate the molecular mechanisms underlying the interaction between Ybt and NLRP3, as the pathogenicity of Ybt may depend on this association.

## 4. Materials and Methods

### 4.1. Cells and Bacterial Strains

ZB-1 is a veterinary clinical pathogenic strain that was isolated from the diarrhea feces of piglets in Yunnan, China. It belongs to the category of enteropathogenic *Escherichia coli* (EPEC) and possesses the HPI (high-pathogenicity island) along with the ability to produce Yersiniabactin [44]. On the other hand, ZB-1Δ*irp2* is a mutant strain derived from ZB-1. In this mutant strain, the *irp2* gene has been deleted, resulting in the loss of yersiniabactin production [45]. Additionally, the IPEC-J2 cells are immortalized small intestinal epithelial cells that have been derived from porcine origin (Guangzhou Jennio Biotech, Guangzhou, China). IPEC-J2 cells are an appropriate in vitro model for simulating *E. coli* infection in animals, as they produce glycocalyx-bound mucus proteins, cytokines, and chemokines. These cells also express Toll-like receptors, allowing them to mimic the natural state of E. coli-infected animals [46].

All *E. coli* ZB-1 and ZB-1Δ*irp2* strains were grown in LB at 37 °C. Small intestinal epithelial cells (IPEC-J2) were cultivated in an essential medium (DMEM) supplemented with 10% fetal bovine serum (FBS), penicillin, and streptomycin. The cells were grown under typical circumstances (37 °C and 5% CO_2_) in tissue culture bottles.

### 4.2. Mice Infection Model and Treatment

Male Kunming mice, aged 5 weeks and weighing 24 ± 2 g, were procured from the Laboratory Animal Center of Kunming Medical University and housed in the Animal Pathology Laboratory of Yunnan Agricultural University. The mice were housed in pathogen-free cages and provided with ad libitum access to food and water under controlled conditions of temperature (25 ± 2 °C), humidity (40 ± 2%), and a 12 h light/dark cycle. The experimental program was approved by the Animal Ethics Committee of Yunnan Agriculture University.

The effect of LPS, which has been reported to cause inflammation and accelerate pyroptosis, was set as a positive control [33,47]. The mice were randomly divided into four groups with 15 mice in each group: control group, ZB-1 group, ZB-1Δ*irp2* group, and LPS group. Bacterial fluid was injected intraperitoneally into the mice to induce infection, with an infection concentration of 1 × 10^7^ CFU/mL and 0.2 mL/head. The control group received only LB via intraperitoneal injection. Body temperature readings were recorded, and clinical signs were evaluated and scored as previously described [48]. A score of 0 indicated a normal reaction to stimuli, while a score of 1 represented a ruffled coat and a delayed response. A score of 2 indicated a response only to repeated stimuli, and a score of 3 indicated an inability to respond or circle in place. If the mice displayed significant lethargy or neurological signs (score = 3), they were euthanized in a humane manner. A score of 4 indicated that the mouse had died.

### 4.3. IPEC-J2 Infected with E. coli

For infection, 2 × 10^6^ IPEC-J2 cells were seeded in 6-well plates and 2 mL of cell culture media (without antibiotics) containing 1 mL of the *E. coli* (OD600 = 0.6) bacterial fluid was added. The experiment consisted of four groups: control, ZB-1, ZB-1Δ*irp2*, and LPS. Following infection, the cells and cell supernatants were collected for further analysis.

### 4.4. ELISA Assay

To determine the presence of IL-1β, IL-18, SOD, sIgA, and LDH in blood or tissue samples, an enzyme-linked immunosorbent assay (ELISA) was conducted using Jiancheng Bioengineering kits from Nanjing, China. The manufacturer’s protocols were followed for all procedures.

### 4.5. Organ Index

At 24 h post-infection, the mice in each group were weighed (while having free access to drinking water). The spleens were aseptically removed and cleaned of surface fat and mesangium before being weighed using an electronic balance and recorded. The organ index was calculated using the formula: organ index (mg·g^−1^) = organ weight (mg) divided by the live weight of the mouse (g).

### 4.6. Histopathology

Mouse tissues were fixed in paraformaldehyde and then subjected to a series of dehydration steps using alcohol–xylene, followed by embedding in paraffin. Tissue sections measuring 5 μm in thickness were obtained using a Leica 2235 microtome from Leica, Wetzlar, Germany, dried at 37 °C, deparaffinized, rehydrated with a series of alcohol–xylene solutions, and washed with deionized water. The sections were then stained with hematoxylin and eosin (H&E) for further analysis.

### 4.7. Transmission Electron Microscopy

Mouse tissues were fixed in glutaraldehyde, washed with PBS, and subsequently fixed in osmium tetroxide. After dehydration using an acetone gradient, the tissues were embedded in Araldite M (Sigma-Aldrich, St. Louis, MO, USA). Ultrathin sections were obtained using an ultramicrotome (Leica, Wetzlar, Germany) and then stained with uranyl acetate and lead citrate. Finally, the sections were examined using a transmission electron microscope (H-7700, Hitachi, Tokyo, Japan) for further analysis.

### 4.8. qRT-PCR and Western Blot Analysis

We conducted specific assays for qRT-PCR and Western blot as previously described [48]. The total RNA was extracted from frozen intestine tissue or harvested cultured cells using the RNAiso Plus Kit from Takara (Dalian, China), and reverse transcription was performed using the PrimeScript RT Master Mix Kit from Takara (Dalian, China) for cDNA synthesis. Quantitative real-time polymerase chain reaction was performed using the SYBR green fast qPCR mix SYBR (Takara, Dalian, China) with the CFX96TM (Bio-Rad, Hercules, CA, USA) real-time fluorescent quantitative PCR system. The mRNA expression was normalized using the housekeeping gene β-actin and the ΔΔCt method. Appendix A contains the primer sequences.

Protein extraction from intestinal tissue was performed using the Thermo #78510 Protein Extraction Kit. The extracted proteins were then separated using SDS-PAGE and transferred onto polyvinylidene fluoride (PVDF) membranes. The PVDF membranes were probed with primary antibodies specific to NLRP3 (#ab270449) and GSDMD (#ab209845), with anti-β-actin (Sigma-Aldrich, St. Louis, MO, USA) used as a loading control for these proteins.

### 4.9. TUNEL Assay

The In Situ Cell Death Assay Kit (Roche, Tucson, AZ, USA) was used to detect cellular DNA fragmentation in intestinal tissue sections, following the manufacturer’s protocol.

### 4.10. NLRP3 siRNA

The siRNA targeting porcine NLRP3 with the specific target sequence 5′-GCAUCUAUUCUGCAAGCUATT-3′ and negative control siRNA (siNC) were designed and synthesized by Sangon Biotech Co., Ltd. (Shanghai, China). Next, 5.0 × 10^5^ cells per well were seeded in 12-well plates, and the cells were transfected with NLRP3 siRNA or control NC by electrotransfection to form the NLRP3 siRNA group and siNC group, respectively. The expression of NLRP3 was measured using qPCR/WB at 24 h intervals.

### 4.11. Immunohistochemistry

We conducted the immunohistochemistry staining of tissue sections using the UltraSensitive^TM^ SP (mouse/rabbit) IHC Kit (MXB, Fuzhou, China), with specific primary antibodies including IL-1β-Abs (126104), IL-18-Absin (125418), NLRP3-Absin (14F468), Caspase-1-Santa Cruz (14F468), and GSDMD-Abcam (ab219800). The results were quantified using ImageJ 1.8 software.

### 4.12. Immunofluorescence Assay (IF) and PI Staining

After pretreatment, IPEC-J2 cells were fixed on glass coverslips using paraformaldehyde, permeabilized with 0.1% Triton X-100 for 10 min, and blocked with 5% bovine serum albumin for 1 h. Specific primary antibodies (NLRP3 and caspase-1) were then incubated overnight at 4 °C. The following day, fluorescent secondary antibodies were incubated for 1 h at room temperature, and the cells were stained with DAPI for 5 min. For PI staining, the cells were seeded in 6-well plates (2 × 10^6^/well), fixed with 4% paraformaldehyde, and incubated with 100 μL of 6.7 μg/mL PI staining solution for 20 min at 37 °C. The stained cells were observed using an Olympus fluorescence microscope, and the results were quantified using ImageJ 1.8 software.

### 4.13. Bioinformatic Analysis

The dataset was obtained from the GEO Database (GSE124917; GEO DataSets—NCBI available at nih.gov (accessed on 17 January 2023)) in the MINiML format. mRNA differential expression was analyzed using the limma package in R software (version: 4.2.2), and adjusted *p* values were used to correct for false-positive results. A threshold of adjusted *p* < 0.05 and log2 (fold change) > 1 or log2 (fold change) < −1 was applied to screen for differentially expressed mRNAs. KEGG enrichment analysis was utilized to determine gene function and high-level genome function information. The ClusterProfiler package 4.0 in R was used to further analyze the function of potential mRNAs and the enrichment of KEGG pathways [49], in order to gain a better understanding of the role of target genes.

### 4.14. Molecular Docking Experiment

The three-dimensional structures of NLRP3 were obtained from the Protein Data Bank RCSB (PDB: 7vtq), while those of Ybt were acquired from PubChem CID (CID: 443589). Molecular docking was performed using CB-Dock2 (available at CB-Dock2 (labshare.cn), accessed on 17 January 2023) after determining the two molecules’ three-dimensional structures [50].

### 4.15. Drug Molecular Screening

We utilized the online tool DrugRep (available at DrugRep (labshare.cn), accessed on 17 January 2023) to screen for drugs with high binding activity to the 3D structure of NLRP3 using the receptor-based screen option. The approved drug library was used for screening to identify potential drug candidates [51].

### 4.16. Statistical Analysis

The experiments were conducted in triplicates and repeated at least three times. Statistical analyses were performed using GraphPad Prism 9.0 (GraphPad Software, San Diego, CA, USA). Student’s *t*-test or two-way ANOVA was used for data analysis, and the results are presented as mean ± SD. A *p*-value less than 0.05 was considered statistically significant.

## Figures and Tables

**Figure 1 ijms-24-11451-f001:**
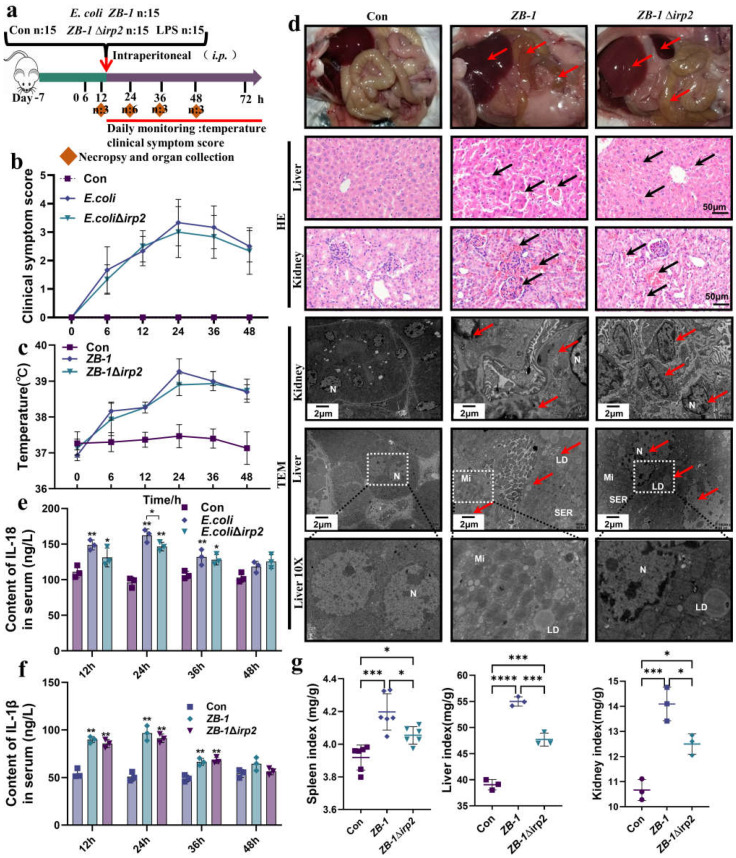
Ybt promotes systemic infection caused by *E.coli*: (**a**) a schematic diagram of the mouse infection model; (**b**) clinical scores of mice after infection; (**c**) changes in body temperature at different time points after infection; (**d**) anatomical changes in infected mice, including pathological changes in the liver and kidney (typical injuries indicated by black arrows, scale bar, 50 μm), and transmission electron microscope observations of *E. coli*-induced cell damage (typical injuries indicated by red arrows; N: nucleus, LD: lipid droplet, Mi: mitochondria, SER: smooth endoplasmic reticulum; the white square box indicates the zoomed-in location); (**e**,**f**) changes in inflammatory factors, including IL-18 and IL-1β, in the serum of infected mice (*n* = 3); (**g**) visceral index of infected mice (including the spleen, kidney, and liver). All data are presented as mean ± SD. **** *p <* 0.0001, *** *p <* 0.001, ** *p* < 0.01, and * *p* < 0.05.

**Figure 2 ijms-24-11451-f002:**
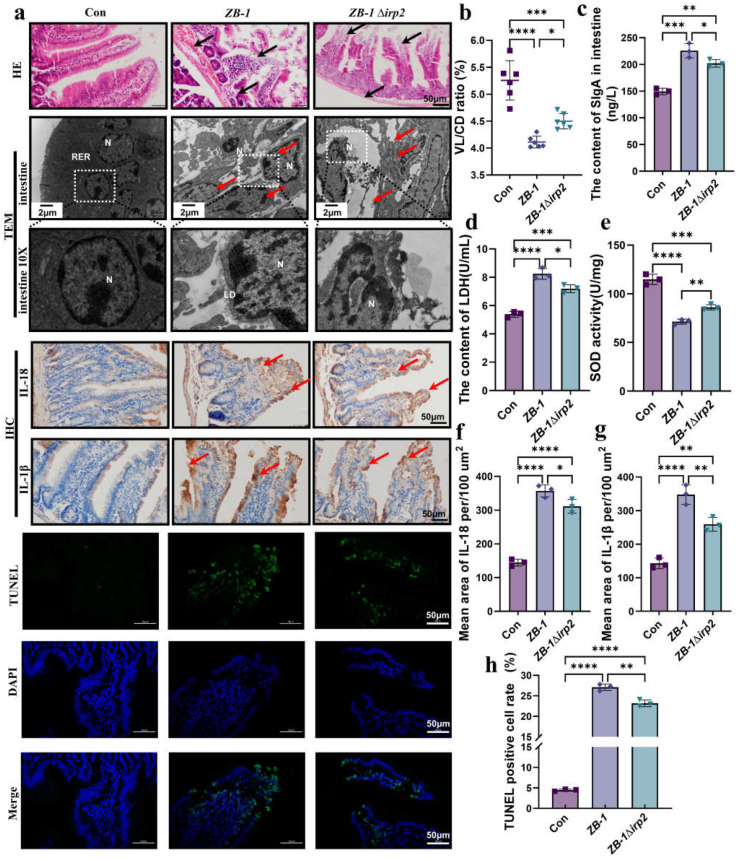
Ybt promotes intestinal inflammation induced by *E.coli*: (**a**) pathological changes in the small intestine in infected mice (scale bar, 50 μm). Transmission electron microscopy observation of *E.coli* damage to cells (N: nucleus, LD: lipid droplet, RER: rough endoplasmic reticulum; the local zoom location is indicated by the white square box). Representative photomicrographs of immunohistochemistry results showing the staining of IL-18 and IL-1β in different groups, and TUNEL staining (scale bar, 50 μm; typical injuries are depicted by the arrows) (*n* = 3); (**b**) the intestine’s length of villi/depth of crypts (VL/CD) was determined (*n* = 3); (**c**,**d**) the content of SIgA and LDH in intestine; (**e**) the activity of SOD in the intestine (*n* = 3); (**f**,**g**) the intestine’ mean density of IL-1β and IL-18 was determined (*n* =3); (**h**) semi-quantification of TUNEL-positive cells in the intestine (*n* = 3). All data are shown as the mean ± SD. **** *p <* 0.0001, *** *p <* 0.001, ** *p <* 0.01, and * *p <* 0.05.

**Figure 3 ijms-24-11451-f003:**
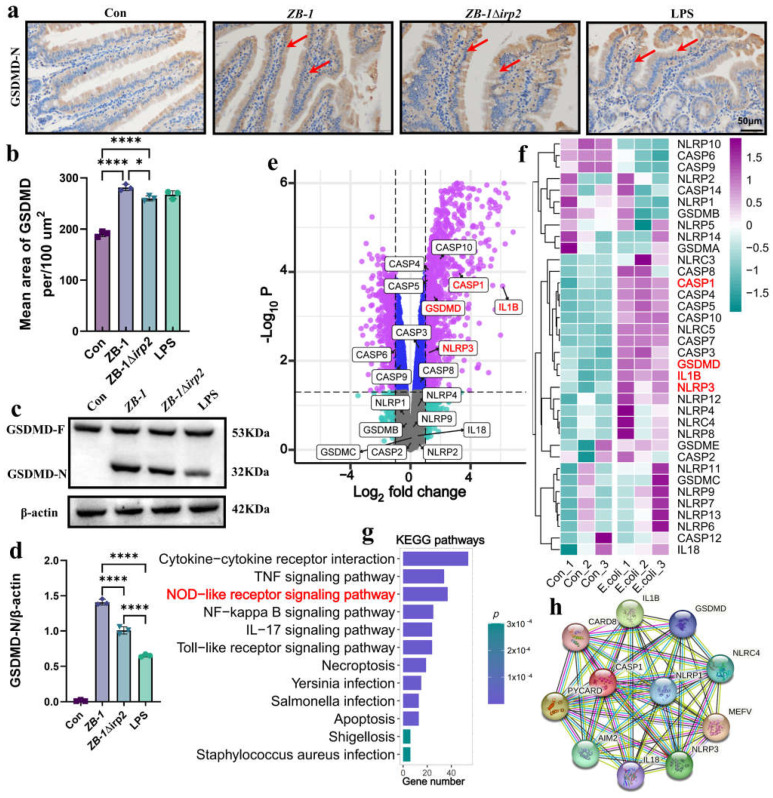
The NLRP3 pathway is involved in *E. coli* infection: (**a**,**b**) representative photomicrographs of immunohistochemistry results showing the staining of GSDMD-N in different groups (scale bar, 50 μm; typical injuries are depicted by the arrows); the intestine’ mean density of GSDMD-N was determined (*n* = 3); (**c**,**d**) Western blotting was used to detect the expression levels of GSDMD and GSDMD-N in the intestine of mice and compared with β-actin (*n* = 3); (**e**) volcano plot: the volcano plot was constructed using the fold change values and *p*-adjusted values. Purple dots represent genes with significant fold change value and *p* value, blue dots represent genes with significant *p* value, gray dots represent genes with insignificant fold change value and *p* value, and green dots represent genes with significant fold change value (The red font highlights genes in the NLRP3 pathway); (**f**) the heatmap of the differential gene expression, with different colors representing the trend of gene expression in different tissues (The red font highlights genes in the NLRP3 pathway); (**g**) functional enrichment: the enriched KEGG signaling pathways were selected to demonstrate the primary biological actions of major potential mRNA (The red font highlights genes in the NOD-like receptor pathway); (**h**) network of interacting proteins predicted via NLRP3. All data are shown as the mean ± SD. **** *p <* 0.0001, and * *p <* 0.05.

**Figure 4 ijms-24-11451-f004:**
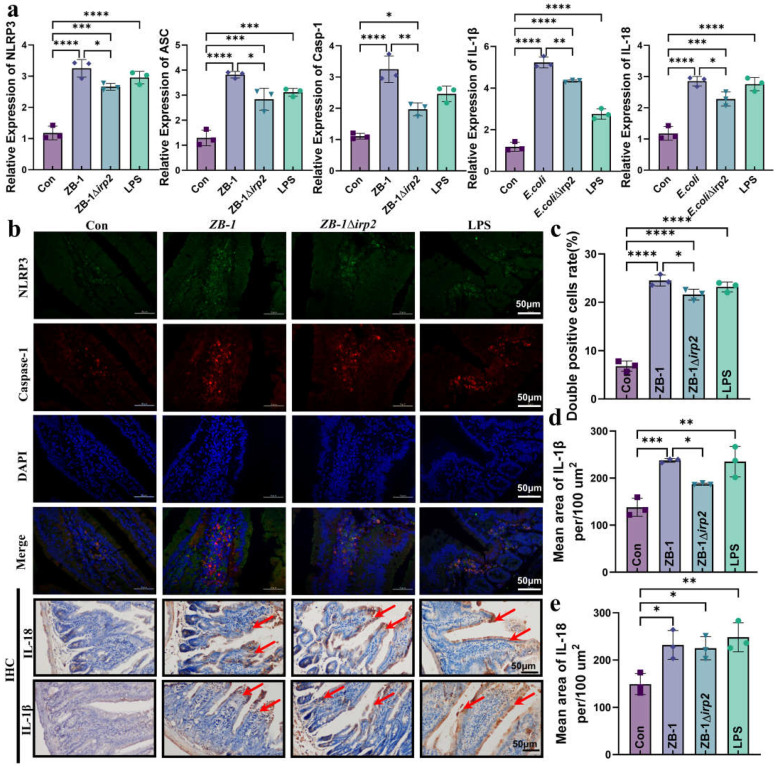
*E. coli*-Ybt induces pyroptosis via the NLRP3 pathway: (**a**) the relative expression levels of NLRP3, ASC, caspase-1, IL-1β, and IL-18 were quantified using qRT-PCR in kidney specimens from mice (*n =* 3); (**b**) immunofluorescence staining was performed to determine the expression localization of NLRP3 and caspase-1 in the intestine. Representative photomicrographs of immunohistochemistry results showing the staining of IL-18 and IL-1β in different groups (scale bar, 50 μm; typical injuries are depicted by the arrows) (*n* = 3); (**c**) the proportion of NLRP3 and caspase-1 double-positive cells in the intestine; (**d**,**e**) the mean density of IL-1β and IL-18 in the intestine was determined (*n* = 3). All data are presented as mean ± SD. **** *p <* 0.0001, *** *p <* 0.001, ** *p <* 0.01, and * *p* < 0.05.

**Figure 5 ijms-24-11451-f005:**
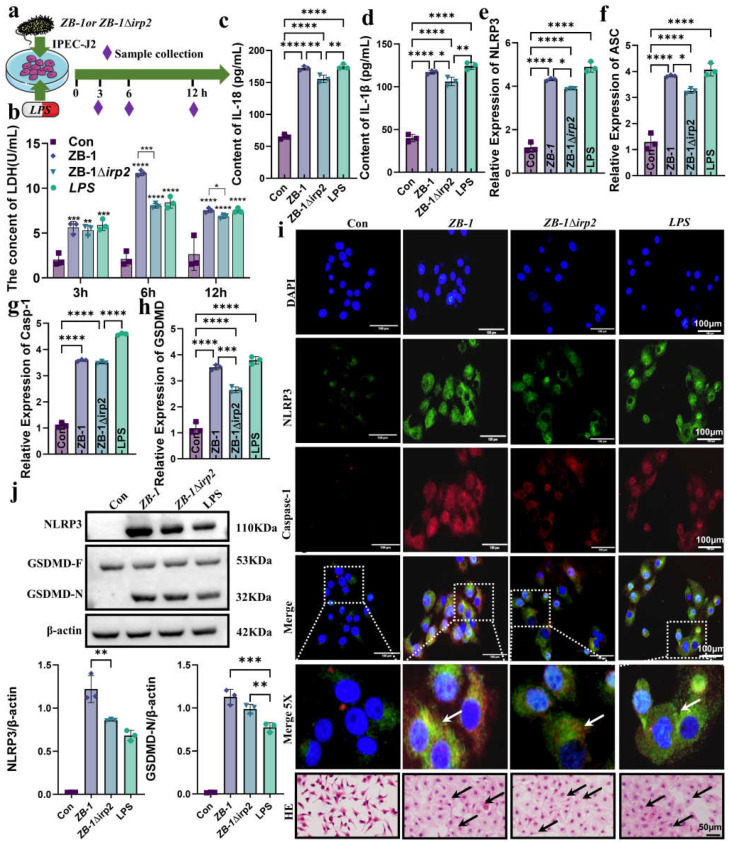
*E.coli*-Ybt induces pyroptosis of small intestinal epithelial cells: (**a**) schematic diagram of IPEC-J2 cell infection model; (**b**–**d**) changes in inflammatory factors in the cell supernatant, including LDH, IL-18, and IL-1β (*n* = 3); (**e**–**h**) the relative expression levels of NLRP3, ASC, caspase-1, and GSDMD were determined using qRT-PCR in IPEC-J2 cells (*n* = 3); (**i**) the assembly of NLRP3 and caspase-1 was observed via laser confocal microscopy (scale bar, 100 μm; the local zoom location is indicated by the white square box, typical inflammasome is represented by a white arrow) (*n* = 3). IPEC-J2 cells were observed via HE staining (scale bar, 50 μm; typical injuries are depicted by the black arrows); (**j**) Western blotting was used to detect the expression levels of NLRP3, GSDMD, and GSDMD-N in IPEC-J2 cells in different treatment groups and compared with β-actin (*n* = 3). All data are shown as the mean ± SD. **** *p <* 0.0001, *** *p <* 0.001, ** *p <* 0.01, and * *p* < 0.05.

**Figure 6 ijms-24-11451-f006:**
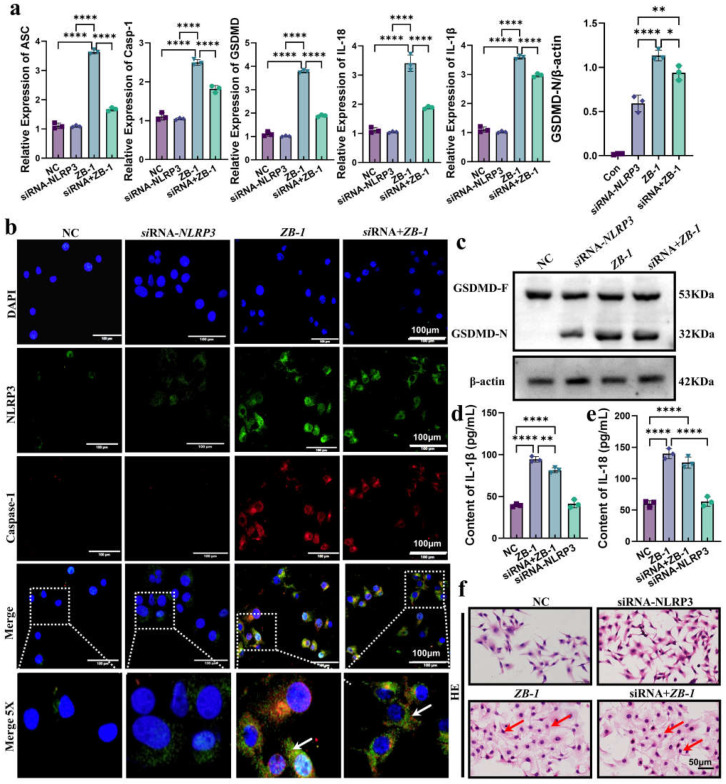
Knockdown of NLRP3 alleviates *E.coli*-Ybt-induced pyroptosis: (**a**) the relative expression levels of *ASC*, *GSDMD*, *caspase-1*, *IL-18*, and *IL-1β* were determined using qRT-PCR in both IPEC-J2 cells and after NLRP3 knockdown (*n* = 3); (**b**) the assembly of NLRP3 and caspase-1 was observed via laser confocal microscopy in both IPEC-J2 cells and after NLRP3 knockdown (scale bar, 100 μm; the local zoom location is indicated by the white square box, typical inflammasome is represented by a white arrow) (*n* = 3); (**c**) Western blotting was used to detect the expression levels of GSDMD and GSDMD-N in IPEC-J2 cells in different treatment groups and compared with β-actin (*n* = 3); (**d**,**e**) changes in inflammatory factors in the cell supernatant, including IL-18 and IL-1β (*n* = 3); (**f**) IPEC-J2 cells were observed via HE staining (scale bar, 50 μm; typical injuries are depicted by the red arrows). All data are shown as the mean ± SD. **** *p <* 0.0001, ** *p <* 0.01, and * *p <* 0.05.

**Figure 7 ijms-24-11451-f007:**
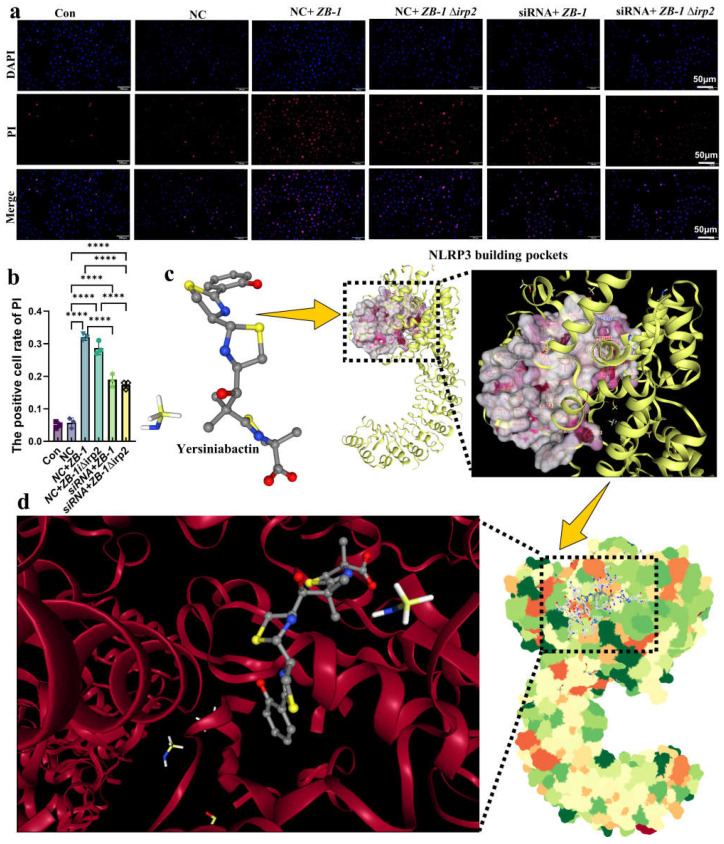
Ybt binds to NLRP3 to induce pyroptosis: (**a**,**b**) after *E. coli* infection, the cells of both IPEC-J2 cells and the NLRP3 knockdown group were stained with propidium iodide (PI) (scale bar, 50 μm). The positive cell rate of PI staining was then determined. (*n* = 3); (**c**) the 3D structure of siderophore Ybt and NLRP3 building pockets; (**d**) the optimal binding structure of Ybt and NLRP3 protein with the local magnification of the binding site. All data are shown as the mean ± SD. **** *p <* 0.0001.

**Figure 8 ijms-24-11451-f008:**
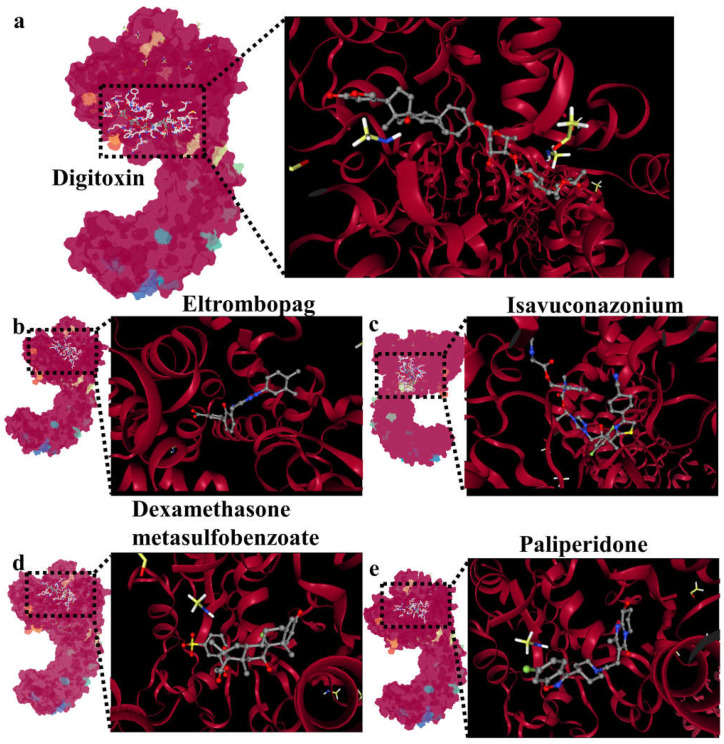
Screening of drug libraries targeting NLRP3: (**a**) molecular docking of digitoxin and NLRP3; (**b**) molecular docking of eltrombopag and NLRP3; (**c**) molecular docking of isavuconazonium and NLRP3; (**d**) molecular docking of dexamethasone metasulfobenzoate and NLRP3; (**e**) molecular docking of paliperidone and NLRP3.

**Figure 9 ijms-24-11451-f009:**
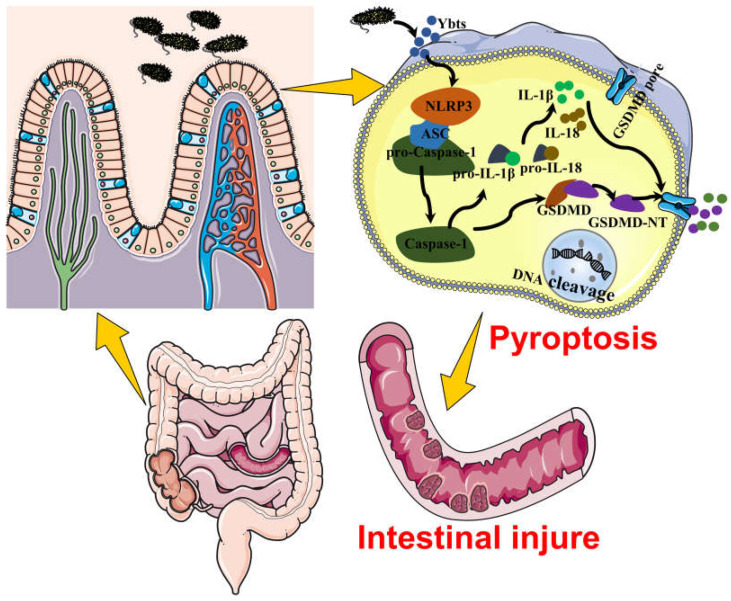
This figure demonstrates that Ybt exacerbates the susceptibility to *E. coli*-induced intestinal inflammation. In addition, intestinal epithelial cell pyroptosis represents another mechanism of intestinal inflammation triggered by *E. coli* infection in mice, and Ybt can induce intestinal injury by promoting pyroptosis.

## Data Availability

The data presented in this study are available in the article.

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
