# Peer review of "Yersiniabactin-Producing E. coli Induces the Pyroptosis of Intestinal Epithelial Cells via the NLRP3 Pathway and Promotes Gut Inflammation"

_ijms, 2023, doi:10.3390/ijms241411451_

Round 1

Reviewer 1 Report

This is an interesting article clarifying the pathogenesis of E. coli infection. I would recommend adding a study flowchart indicating how many animals were in groups and at what stages and how they were examined.

Reviewer 2 Report

The manuscript of Wang et al., 2023  describes how Yersinibactin-producing E.coli induces pyroptosis of intestinal-epithelial cells which promotes gut inflammation through NLRP3 pathway, for which they claim it is a new by E. Coli promotes gut inflammation.

Since in the beginning it is mention that E.coli-Yersinibactin is competing for Fe3+ in the host, in the early stage after infection before intestine comes into pyroptosis the level of Fe3+ in the intestine could be measured.

Assay that are done with E.coli ZB-1 and E.coli and ZB-1Δirp2  to me they do not so significantly different, why there is so many assays done with this both strain?

The assay of pyroptosis as type of cell death is proofed with TUNEL staining, that show DNA fragmentation were there done also some other assay of cell death like staining with Pi or to exclude apoptosis staining  for caspase-3 and annexin staining?

Reviewer 3 Report

Manuscript 2450810: Yersiniabactin-Producing E. coli induces pyroptosis of intestinal epithelial cells via the NLRP3 pathway and promotes gut inflammation

 This manuscript suggest that infection by Yersiniabactin (Ybt)-Producing E. coli induces pyroptosis and exacerbates gut inflammation during systemic infection in mice. The mechanism involve the assembly and activation of the NLRP3, leading to a functional cleavage of GSDMD that resulted in pyroptosis of intestinal epithelial cells, and consequently gut inflammation.

All experiments showed good quality and proper controls. However, important information and references are missing. For that reason, this paper needs major modifications.

 Concerns and suggestions are listed below:

 Material and methods indicate that 15 mice in each group were infected. But the quantitative graphs and figure legends showed n= 3. Since authors have higher number of infected mice, I strongly suggest to increase the number of mice in the quantification.   

 In addition to table S1, the text should provide a general description (strain, collection and/or isolation, laboratory that provided you this strain) of E. coli used in this study. Provide evidence of irp2 gene deletion or indicate a reference with this information.

 Images need to be reorder. Please add images in the correct order and next to each image the respective quantification.

 Describe the clinical symptoms and the score numbers that were used for measuring index of injury.

 Authors must indicate the aim of measuring secretory IgA, LDH and SOD activity; and what the results represent.

 Authors should add a brief explanation about selecting IPEC-J2 cells for in vitro experiments.

Visceral index is described for spleen only. Is it the same for Liver and Kidney?

Add information about the siRNA targeting NLRP3 (company, design and/or reference).

 For IHC, IF and PI staining quantification, please indicate the number of images taken per samples.

 Clarify if the cell infection was performed in medium with or without antibiotics.

 Figure legends do not indicate all the P values showed in each figure. Please add all P values.

 Add scale bar in transmission electron microscope images

 Indicate in the text and figure legend that Knockdown of NLRP3 was performed in IPEC-J2 cells.

 Figure 6C is not described in the text. And figure legend indicate that is a WB of gut mice. Is this correct?

 Figures 2B, 2F, 2G, 3B, 4B and 4C do not indicate units. Use um to indicate the area.

 Figure legend 5J indicate WB of kidney and text indicate WB of cells. Which one is correct?

 Line 207 indicate figure 4e, Is this correct?

Minor editing of English language required. There are some typos in the text.

Reviewer 4 Report

In this article entitled “Yersiniabactin-Producing E. coli induces pyroptosis of intestinal epithelial cells via the NLRP3 pathway and promotes gut inflammation” authors have discovered a new role for Yersiniabactin (Ybt) in E. coli pathogenesis. Authors show that Ybt activates the NLRP3 infllammasome to induce pyroptosis in intestinal epithelial cells for gut inflammation. I have several concerns regarding the experimental design and findings.

Major points:

1.     In the abstract authors mention “Pyroptosis is a programmed cell death mediated by 24 Gasdermin family proteins found in macrophages”. Please correct this as GSDMD protein is ubiquitously present in various immune and non-immune cells. Even the inflammasome activation and GSDMD cleavage occurs in various immune and non-immune cells as shown by Kumari et al. Cell Reports, Volume 35, Issue 320 April 2021, 109012

2.     In the introduction, authors have written one sentence “The NLRP3/Caspase-1 pathway is the typical pyroptosis pathway” which in my opinion is misleading in current era as various other inflammasomes have been researched and their roles in health and disease have been established.

3.     In Fig 1c, authors are showing hyperthermia, however, previous literature suggest hypothermia upon E. coli infection to mice. Please explain the results in Fig 1c.

4.     Authors claim that SOD activity is decreased and hence DNA fragmentation is increased in ZB-1 strain infection in comparison to the mutant. Did authors test the involvement of DNA sensor-mediated inflammasome activation such as in the case of Aim2 inflammasome?

5.     In the figure 2, authors claim that Ybt promotes E. coli gut inflammation by inducing DNA fragmentation. Do authors suggest the involvement of more than one inflammasome in the phenotype?

6.     In Fig 3C, authors have shown the cleavage of GSDMD upon Ybt-E.coli infection in parallel comparison to LPS stimulated mice intestine. However the cleavage of GSDMD is not significantly different among the groups. Please clarify whether the GSDMD cleavage experiment was done in germ-free mice, as the conventionally raised mice (specific pathogen free mice) show self-cleavage of GSDMD in the intestine (refer this papre Zhang et al., Sci. Immunol. 7, eabk2092 (2022) 4 February 2022). Please mention what portion of the intestine (ileum, caecum, colon) is taken for the experiments. Please correct the size of GSDMD in Fig 3C.

7.     Fig 3h: In this protein network, proteins such as Aim2, NLRC4 and Casp8 could have independent roles in activating inflammasome. However, authors  have not addressed the potential roles of these proteins, rather authors are emphasizing the role of just NLRP3. Please explain this? Additionally, in the Fig 3f, which is a human dataset (not mouse dataset) it is biased to say that only NLRs are involved as Casp-4 is also shown to be highly expressed in that dataset, which is the key enzyme in human cells to activate inflammasome upon cytosolic LPS sensing. Indeed, in the Fig 3a, 3c, authors have shown the role of LPS in GSDMD cleavage in intestine, which clearly defines the role of Casp11/4 in intestinal GSDMD cleavage. So, the exclusive role of NLRP3 upon ZB1-E.coli infection is not convincing to me. authors are suggested to use NLRP3 deficient system to prove their hypothesis. Overall, using LPS as positive control is not correct as LPS injection to mice causes the activation of casp11-mediated non-canonical inflammasome. In case of whole bacterial infection, LPS could still be involved in non-canonical inflammasome activation, which authors have neglected to discuss. it is suggested to use hypovesiculating mutant of ZB1-E.coli bacterial strain which is expected to secrete less OMV containing LPS, to rule out the involvement of LPS-mediated pyroptosis during infection with ZB-1 E. coli.

8.     It will be nice to see whether Ytb causes intestinal inflammation in in vivo animal model.

Minor editing on language is required
